# The frequency limit of outer hair cell motility measured in vivo

**Anna Vavakou, Nigel P Cooper, Marcel van der Heijden\***

Department of Neuroscience, Erasmus MC, Rotterdam, Netherlands

**Abstract** Outer hair cells (OHCs) in the mammalian ear exhibit electromotility, electrically driven somatic length changes that are thought to mechanically amplify sound-evoked vibrations. For this amplification to work, OHCs must respond to sounds on a cycle-by-cycle basis even at frequencies that exceed the low-pass corner frequency of their cell membranes. Using in vivo optical vibrometry we tested this theory by measuring sound-evoked motility in the 13–25 kHz region of the gerbil cochlea. OHC vibrations were strongly rectified, and motility exhibited first-order low-pass characteristics with corner frequencies around 3 kHz– more than 2.5 octaves below the frequencies the OHCs are expected to amplify. These observations lead us to suggest that the OHCs operate more like the envelope detectors in a classical gain-control scheme than like high-frequency sound amplifiers. These findings call for a fundamental reconsideration of the role of the OHCs in cochlear function and the causes of cochlear hearing loss.
DOI: https://doi.org/10.7554/eLife.47667.001

## Introduction

The hair bundles of auditory sensory cells are deflected by sound-driven vibrations, causing mechano-electric transduction channels to open and close. The resulting receptor current modulates the cell's membrane potential. The mammalian cochlea contains two distinct types of hair cells. The vast majority of nerve fibers that carry the acoustic information to the brain innervate the inner hair cells (IHC). Up to a few kilohertz, IHC synapses can 'phase-lock,' that is, code the individual cycles of tones. At higher frequencies (>3 kHz), phase-locking rapidly declines and neural coding relies on the DC component of the IHC receptor potential generated by the asymmetric, or rectifying, nature of the IHC receptor current (*Russell and Sellick, 1978*).

Outer hair cells (OHC) modify the mechanical vibrations inside the organ of Corti (OoC), enabling frequency tuning and dynamic range compression. Dysfunctional and missing OHCs strongly reduce sensitivity, and this is a major cause of sensorineural hearing loss (*Ryan and Dallos, 1975*). The discovery of electromotility, length changes of isolated OHCs (*Brownell et al., 1985*) driven by variations in the membrane potential (*Santos-Sacchi and Dilger, 1988*), has intensified the study of OHCs and their functional significance. The membrane protein responsible for electromotility has been identified (*Zheng et al., 2000*), and prestin knockout mice have profound hearing loss (*Liberman et al., 2002*). The dominant view is that OHC electromotility drives vibrations within the OoC in a cycle-by-cycle manner (*Ashmore, 2011*) over the entire audible range, which extends up to 150 kHz in some species (*Vater and Kössl, 2011*). If this view is correct, the AC receptor potentials of OHCs must be large enough to be effective up to high frequencies, even though the membrane capacitance is expected to reduce the AC receptor potentials (and hence the OHCs' motility) at a rate of 6 dB per octave (*Dallos, 1984*). The functional implication of this electrical low-pass filtering is a limitation in the OHC's ability to provide cycle-by-cycle mechanical feedback, known as the RC problem (*Ashmore, 2011*; *Housley and Ashmore, 1992*). The electrical corner frequency of OHCs has been measured electrophysiologically in vitro, with highest values ranging from 480 Hz (*Mammano and Ashmore, 1996*) to 1250 Hz (*Johnson et al., 2011*), but no systematic in vivo data

*For correspondence:
m.vanderheyden@erasmusmc.nl

Competing interests: The authors declare that no competing interests exist.

**eLife digest** Our ears give us our sense of hearing. Their job is to collect sounds and pass this information on to the brain. Hair cells, a special group of cells in the ear, are responsible for detecting sound vibrations and turning them into the electrical signals that our brains can understand.

The ear contains two populations of hair cells: inner hair cells that send signals to the brain, and outer hair cells that act as a protective 'buffer' by modulating sound vibrations entering the innermost part of the ear. When outer hair cells are damaged, the vibrations picked up by inner hair cells are much smaller than in a healthy ear. This has led to the idea that outer hair cells actively amplify sounds before passing them on. That is, outer hair cells simultaneously act like microphones (by receiving sound from the environment) and loudspeakers (by re-emitting magnified vibrations).

One problem with this amplifier theory is that it cannot explain how some animals are able to hear extremely high-pitched sounds. If the theory is true, outer hair cells should be able to re-emit ultrasonic vibrations. However, some observations suggest that they may not vibrate fast enough to do so.

To test the amplifier theory, Vavakou et al. measured how outer hair cells in the ear of Mongolian gerbils responded to different sounds. This revealed that the motion of these cells could keep up with moderately high sounds (around the upper end of a piano's range), but were too sluggish to amplify ultrasound despite gerbils having good ultrasonic hearing. Further experiments showed that instead of acting like amplifiers, outer hair cells seem to monitor the loudness of sound and adjust the level accordingly before passing the vibrations on to the inner hair cells.

These results shed new light on how outer hair cells help our ears work. Since damage to these cells can cause hearing loss, understanding how they work could one day guide new methods of protecting or even restoring hearing in vulnerable patients.

DOI: https://doi.org/10.7554/eLife.47667.002

exist due to technical limitations and the cochlea's extreme vulnerability. In addition to the electrical filtering, the motile process itself may also be too slow to provide high-frequency mechanical feedback. An early in vitro report claiming a bandwidth for electromotility of at least 79 kHz (*Frank et al., 1999*) was recently challenged (*Santos-Sacchi and Tan, 2018*). Again, in vivo estimates of the corner frequency of motility are missing. Here we use optical vibrometry to measure non-linear components of the OHCs' motile response and determine the corner frequency of OHCs in the high-frequency region of the intact gerbil cochlea.

## Results and discussion

In response to a tone pair (*Figure 1A*), vibrations in the OHC/Deiters' cell region showed a strong envelope-following component (*Figure 1B*). This reveals a significant degree of rectification in OHC motion, producing 2nd-order distortions (DP2s) such as the 'quadratic difference tone' at f2-f1. Using multitone stimuli, we mapped the spatial profile of the DP2s inside the OoC by cross-section measurements. DP2s were concentrated in the OHC/Deiters' cell 'hotspot area' (*Cooper et al., 2018*) (*Figure 1C and D*). They were observed at stimulus levels as low as 25 dB SPL (*Figure 1—figure supplement 1*), and disappeared post mortem (*Figure 1—figure supplement 2*). These observations confirm that OHCs are the source of the DP2s long known to exist from psychophysical (*Zwicker, 1979*), electrophysiological (*Kim et al., 1980*; *Nuttall and Dolan, 1993*), and cochlear-mechanical (*Cooper and Rhode, 1997*) studies. The OHC origin of DP2s is consistent with the significant rectified ('DC') component and 2nd harmonics observed in in vivo recordings of OHC receptor potentials (*Dallos, 1986*; *Cody and Russell, 1987*) and cochlear microphonics (*Gibian and Kim, 1982*).

Because rectification by OHC bundles produces DP2s in the receptor current, DP2s are an inevitable part of the electromotile response. More importantly, this part can be isolated through spectral analysis from recorded cochlear vibrations. This makes the DP2 spectrum ideally suited to studying the RC problem. To assess spectral trends, we employed a zwuis tone complex (*Figure 1E*), a stimulus designed to produce a rich DP2 spectrum upon rectification (*van der Heijden and Joris, 2003*;

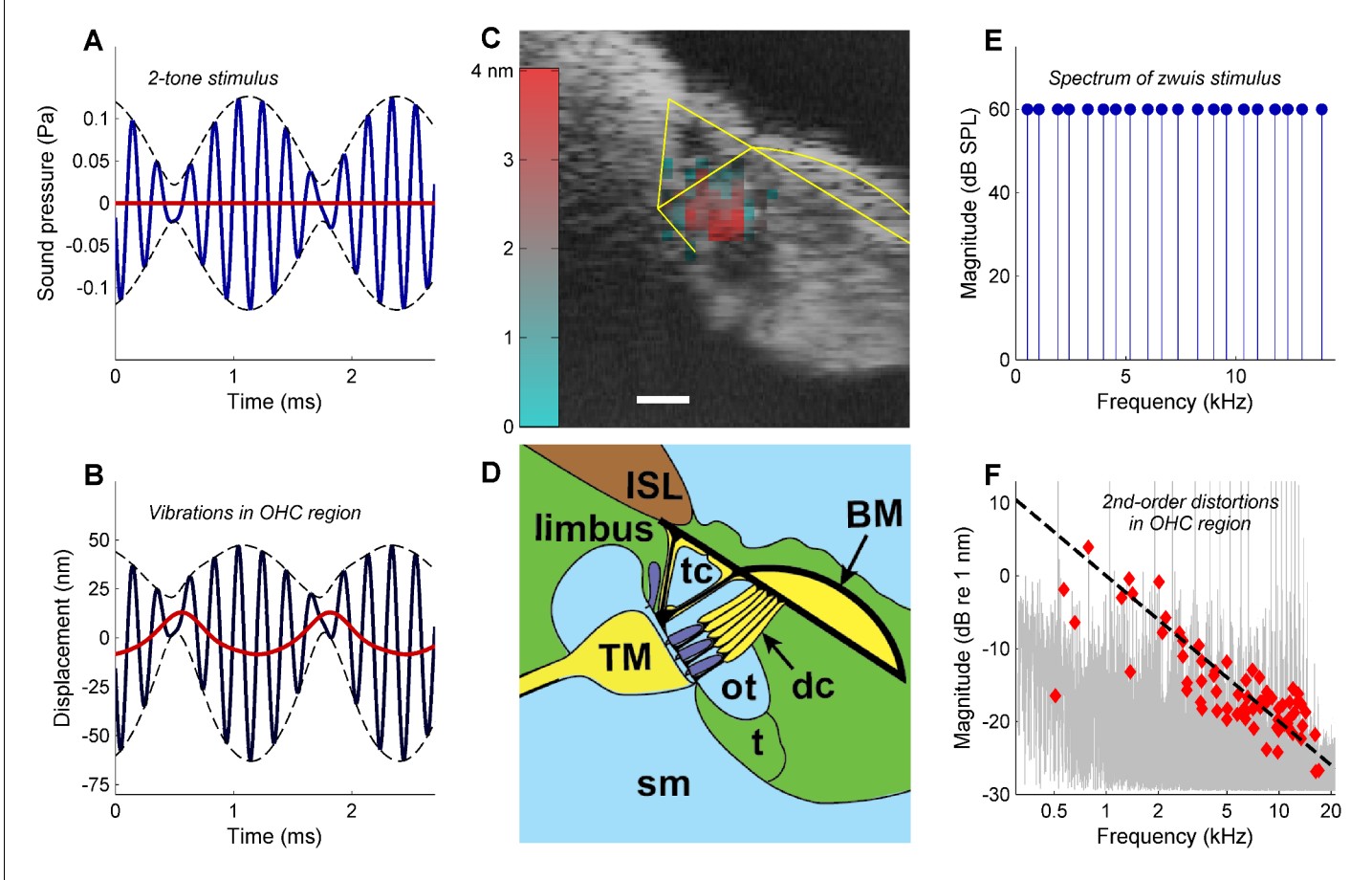

**Figure 1.** Rectification in the mechanical response of OHCs. (**A**) Two-tone stimulus with primary frequencies 4600, 5400 Hz; 70 dB SPL. *Blue line*, waveform; *dashed black lines*, envelope; *red line*, lowpass-filtered waveform (2000-Hz cut-off). (**B**) Mechanical displacement evoked by the two-tone stimulus, recorded in the gerbil OHC/Deiters' cell region (13 kHz CF). *Black line,* displacement waveform; *dashed black lines*, envelope. Rectification is illustrated by the *red line* obtained by low-pass filtering (2000-Hz cut-off). Positive polarity indicates displacement away from the measurement probe, that is vertically downwards in (**C**) and (**D**). (**C**) OCT reflectance image (*grayscale*), with structural framework of Corti's organ (*yellow*) superimposed for reference. Color-coded overlay: total RMS value of $2^{nd}$-order distortion products (DP2s) evoked by a 12-tone complex, 2–12 kHz; 60 dB SPL. DP2s dominate in the OHC region. Scale bar, 0.025 mm. (**D**) Underlying anatomical structures. BM = basilar membrane; ISL = inner spiral lamina; sm = scala media; dc = Deiters' cells; t = tectal cells; TM = tectorial membrane; tc = tunnel of Corti; ot = outer tunnel. Hair cells, *dark blue*. (**E**) Zwuis stimulus (see text). (**F**) Vibration spectrum recorded in OHC region in response to the zwuis stimulus. *Red diamonds*, Rayleigh-significant DP2s. Dashed line, 6-dB/octave roll-off.

DOI: https://doi.org/10.7554/eLife.47667.003

The following source data and figure supplements are available for figure 1:

**Source data 1.** MATLAB binary file containing the data shown in *Figure 1*.
DOI: https://doi.org/10.7554/eLife.47667.006
**Figure supplement 1.** DP2s at low sound intensities.
DOI: https://doi.org/10.7554/eLife.47667.004
**Figure supplement 2.** Post mortem disappearance of DP2s.
DOI: https://doi.org/10.7554/eLife.47667.005

*Victor, 1979*). Rectifying an *N*-component zwuis stimulus generates $N^2$ distinct DP2 components at frequencies $f_k \pm f_m$, each of which can be traced back to a pair of interacting primary frequencies ($f_k$, $f_m$). The vibration spectrum obtained in the OHC region (*Figure 1F*) evoked by this stimulus reveals a rich family of DP2s having a systematic 6-dB/octave roll-off. This roll-off confirms the action of a low-pass filter between the hair bundle's rectification and the motile response. Accurate estimation of the corner frequency, however, is hampered by the ~10-dB scatter of DP2 magnitudes within

each frequency band. We identified three causes of this scatter (*Figure 2*). Their elimination reduces the scatter substantially, paving the way to accurate estimates of OHC corner frequencies.

The first cause of scatter is combinatorial: when supplying an equal-amplitude input to a rectifier, the resulting second harmonics $2f_k$ are 6 dB below the remaining DP2s ($f_k \pm f_m$, $k > m$) (*Figure 2A*). This reflects the binomial coefficients occurring in the second-order terms of the power series describing the nonlinearity (*Meenderink and van der Heijden, 2010*). Since each DP2 component can be uniquely traced back to its 'parent primaries,' this is readily corrected. The second cause of scatter is the spatially distributed nature of DP2 generation. The primaries and DP2s propagate as

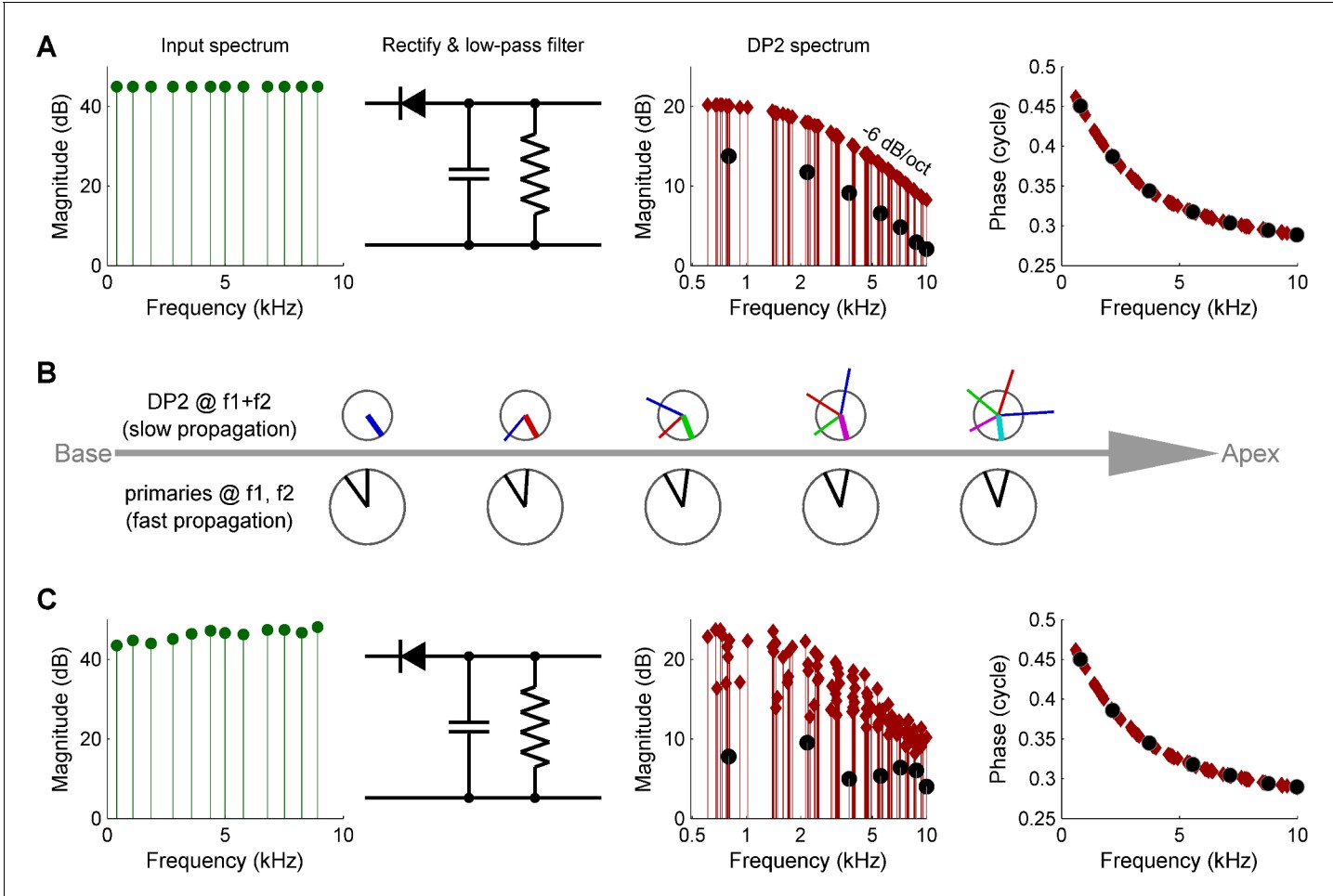

**Figure 2.** Three causes of magnitude scatter in DP2 spectra. (**A**) Combinatorial effect illustrated by feeding an equal-amplitude zwuis tone complex to a nonlinear circuit comprising a rectifier and low-pass filter (corner frequency 2.5 kHz). The 2$^{nd}$ harmonics (*solid circles*) are 6 dB weaker than the remaining DP2s (*red diamonds*). (**B**) Vector addition of DP contributions along the traveling wave (left to right). Lower row of 'clocks' depict amplitude and phase of the primaries $f_1,f_2$ <<CF. They accumulate little phase and their amplitude hardly grows upon traveling. Upper clocks depict a near-CF DP2 at $f_1+f_2$. Colors indicate the origin of each local contribution. Near-CF DP2 components suffer considerable phase accumulation and amplitude variation while traveling. The eventual amplitude (rightmost location) is the vector sum of multiple contributions widely differing in phase and amplitude. This interference obfuscates the spectral properties of DP2 generation investigated here. (**C**) Unequal primary amplitudes entering the nonlinear circuit generate a predictable scatter in DP2 magnitudes (see text). Companion phases are not affected by lack of equalization of the input. The 0.5-cycle low-frequency limit of the phase reflects the 'negative polarity' of the rectification.

DOI: https://doi.org/10.7554/eLife.47667.007

The following figure supplements are available for figure 2:

**Figure supplement 1.** Propagation of DP2s in the 18-25-kHz region.

DOI: https://doi.org/10.7554/eLife.47667.008

**Figure supplement 2.** Comparison of linear response components and second-order distortion products (DP2s).

DOI: https://doi.org/10.7554/eLife.47667.009

traveling waves (*Kim et al., 1980*; *Gibian and Kim, 1982*), (*Figure 2—figure supplement 1*), so the recorded DP2s are a vector sum of contributions along the path from stapes to recording place (*Schroeder, 1969*). The magnitude of slowly propagating components is affected by interference across generation loci (*Figure 2B*), and the growth and subsequent decay of components entering their peak region further obfuscates their original magnitude. These confounding effects of propagation are eliminated by setting an upper frequency limit to the primaries and DP2s used for stimulation, analysis and iterated adjustment of the stimulus as described below. For this frequency limit we choose half the characteristic frequency (CF/2). Wave propagation below CF/2 is too fast to cause interference and magnitudes change little during fast propagation (*Ren et al., 2011*) (*Figure 2—figure supplement 1*).

The third cause of scatter in the DP2 spectrum is the unequal amplitude of the primary components entering the rectifier, that is, the effective input that deflects the OHC bundles. Possible causes of unequal amplitudes at the OHC input include imperfections in the sound calibration as well as non-flatness of middle-ear transfer and intracochlear propagation. Even a perfectly regular trend in the input spectrum such as a roll-off creates a scattered effect in the DP2 spectrum (see Appendix 1). The amplitude of a DP2 component is proportional to the product of its parents' amplitudes (*van der Heijden and Joris, 2003*). This bilinearity causes a scatter in DP2 magnitude (expressed in decibels) equal to twice the range of the primary input magnitudes (*Figure 2C* and *Appendix 1—figure 1*). If the OHC input were known, it would take a simple adjustment of the stimulus spectrum to equalize the primary amplitudes at the OHC input, and thereby regularize the DP2 spectrum as in *Figure 2A*. Current in vivo measurement techniques lack the spatial resolution to determine OHC bundle deflection, but the effective OHC input can be retrieved from the rich DP2 spectrum by exploiting the bilinear relationship between primary and DP2 amplitudes. This computational method was previously used to retrieve the effective IHC input from auditory-nerve responses (*van der Heijden and Joris, 2003*). Here we use it to compute the effective OHC input (see Appendix 1) and to adjust the stimulus accordingly. The OHC input thus obtained differs from the linear component of OHC motion, and in fact resembles basilar membrane motion more closely (*Figure 2—figure supplement 2*). This means that motion recorded in the OHC region may not be used as a proxy for OHC input. The resemblance between basilar membrane motion and computed OHC input may shed light on the mechanisms underlying the deflection of OHC hair bundles. Within the current study, however, the OHC input is primarily of methodological interest.

Experimental equalization of the effective OHC input and compensation for the combinatorial effect had the predicted effect of markedly reducing the scatter in the DP2 spectrum (*Figure 3*). After equalization, the DP2 spectrum recorded from the OHCs closely resembles that of the simple nonlinear circuit of *Figure 2A* with its first-order low-pass characteristics. This holds for both the magnitude and the phase, having a minus 6-dB/octave high-frequency slope and a 0.25-cycle high-frequency asymptote (*Figure 4C*) respectively. The corner frequency was 2.5 kHz, 2.7 octaves below the 16-kHz CF.

We obtained data from different cochlear regions in different animals, with CFs ranging from 13 to 25 kHz. The adjustment of the relative stimulus amplitudes always reduced the scatter of the DP2 magnitudes. First-order low-pass characteristics were consistently found across CFs (*Figure 4*). The corner frequencies obtained ranged from 2.1 to 3.3 kHz, showing a weak trend of increasing with CF (*Figure 4D*).

In summary, the rectification displayed in vivo by OHC motility provides a unique opportunity to directly measure OHC corner frequencies without opening the cochlea. When equalizing the primary amplitudes at the OHC input, the DP2 spectra reveal an unmistakable first-order low-pass character, both in terms of magnitude and phase. In the frequency range probed here (below CF/2) stiffness dominates OoC impedance rendering displacement proportional to force (*Dong and Olson, 2009*). Thus within the framework of models in which OHCs directly push the basilar membrane (e.g., *Ramamoorthy et al., 2007*), electromotile force itself suffers from the 6-dB/octave roll-off.

The corner frequencies of 2.1–3.3 kHz that we measured in vivo were 2.8 ± 0.2 octaves below the CFs of our recording locations. These corner frequencies are higher than values of membrane corner frequency from in vitro studies at lower CF: 480 Hz (guinea pig, CF ~7 kHz) (*Mammano and Ashmore, 1996*); 300–1250 Hz (gerbil, CF,~350–2500 Hz) (*Johnson et al., 2011*). The in vivo data fall considerably short of the extrapolations to higher CFs made in the in vitro gerbil study (*Johnson et al., 2011*) (which predict electrical corner frequencies of 6.5–11 kHz for the CFs tested

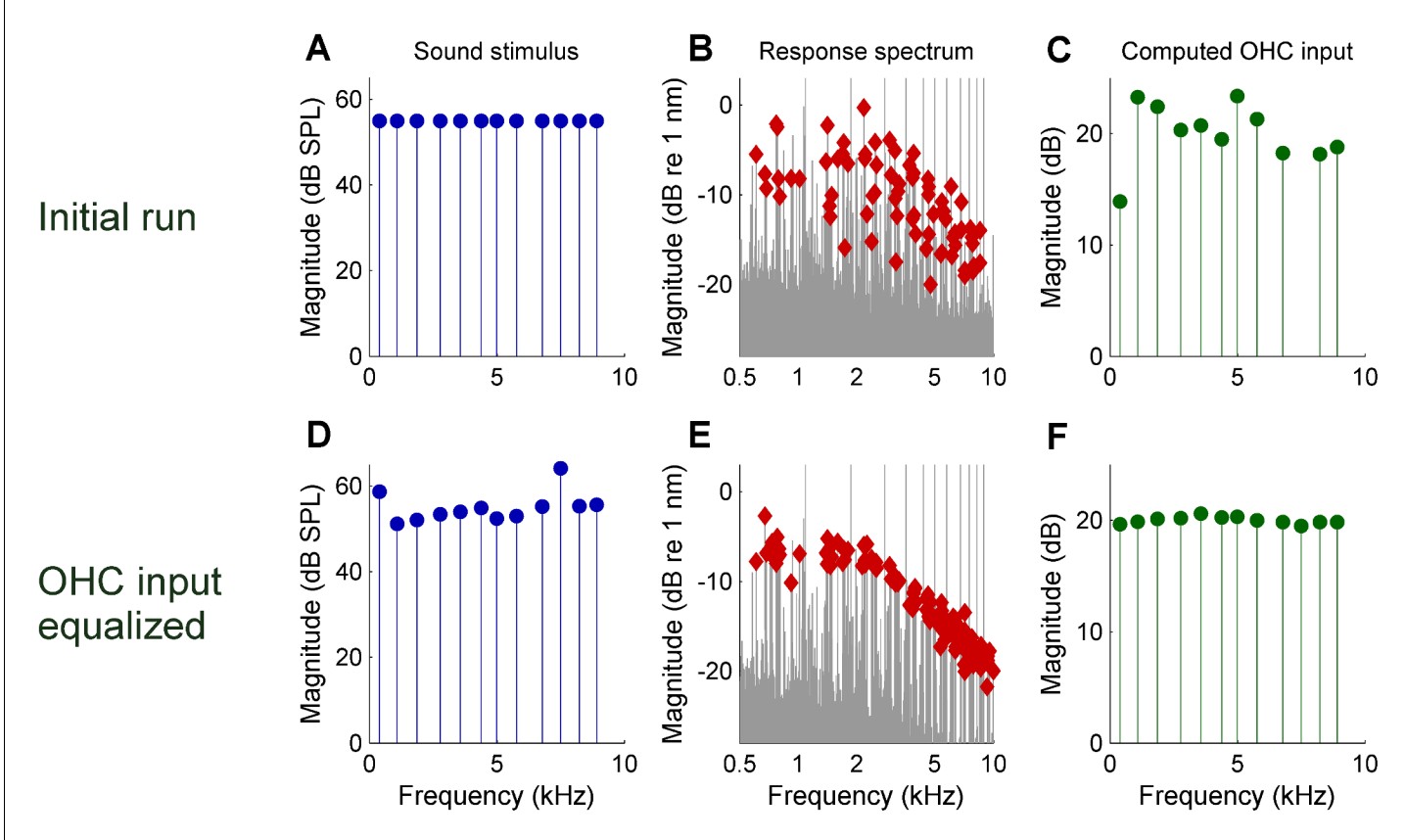

**Figure 3.** Reducing the scatter of DP2 magnitudes by equalizing the effective OHC input during an experiment. (**A**) Spectrum of the initial zwuis acoustic stimulus. (**B**) Spectrum of the vibrations recorded in the OHC region; CF = 16 kHz. Rayleigh-significant DP2s marked as *red diamonds*; 2nd harmonics corrected for the 6-dB combinatorial effect. (**C**) Effective OHC input computed from the DP2 magnitudes. (**D**) Spectrum of the adapted stimulus (4th iteration) aimed at equalizing the effective OHC input. (**E**) Resulting DP2 spectrum. (**F**) Effective OHC input. Note that the equalized input spectrum in F (compared to C) reduces the DP2 scatter in E (compared to B).

DOI: https://doi.org/10.7554/eLife.47667.010

The following source data is available for figure 3:

**Source data 1.** MATLAB binary file containing the data shown in *Figure 3*.

DOI: https://doi.org/10.7554/eLife.47667.011

here). Our observation of a simple 6-dB/octave roll-off and minus 0.25-cycle phase asymptote indicates the dominance of a single low-pass mechanism in the entire frequency range tested. Comparison with the in vitro data suggests that this dominant factor is the RC time of the cell membrane, which is fundamental to the operation of all biological cells. The somewhat higher corner frequencies of the present study (compared to the in vitro data) may be attributed the more basal location of the OHCs of the present study.

A corner frequency at 2.8 octaves below CF implies a 17-dB attenuation of the CF component. The shallow increase of OHC corner frequency with increasing CF suggests an even stronger attenuation at higher CFs than studied here. When driven by a sufficiently large electrical input, OHC motility can generate vibrations up to very high frequencies, both in vitro (*Frank et al., 1999*) and in vivo (*Ren et al., 2016*), but for acoustic stimulation the low-pass filtering of the receptor potential will limit the frequency range. Various schemes (reviewed in *Johnson et al., 2011*) have been proposed to push the frequency limit of electromotility beyond the corner frequency of the OHC cell membrane into the CF range. Our findings do not support such schemes, as the ~2.5-kHz corner frequency is evident in the motile response itself.

We assessed the quantitative effect of low-pass filtering by the OHCs (Appendix 2). A 16-kHz tone at the behavioral threshold of the gerbil is estimated to evoke an AC component of the OHC

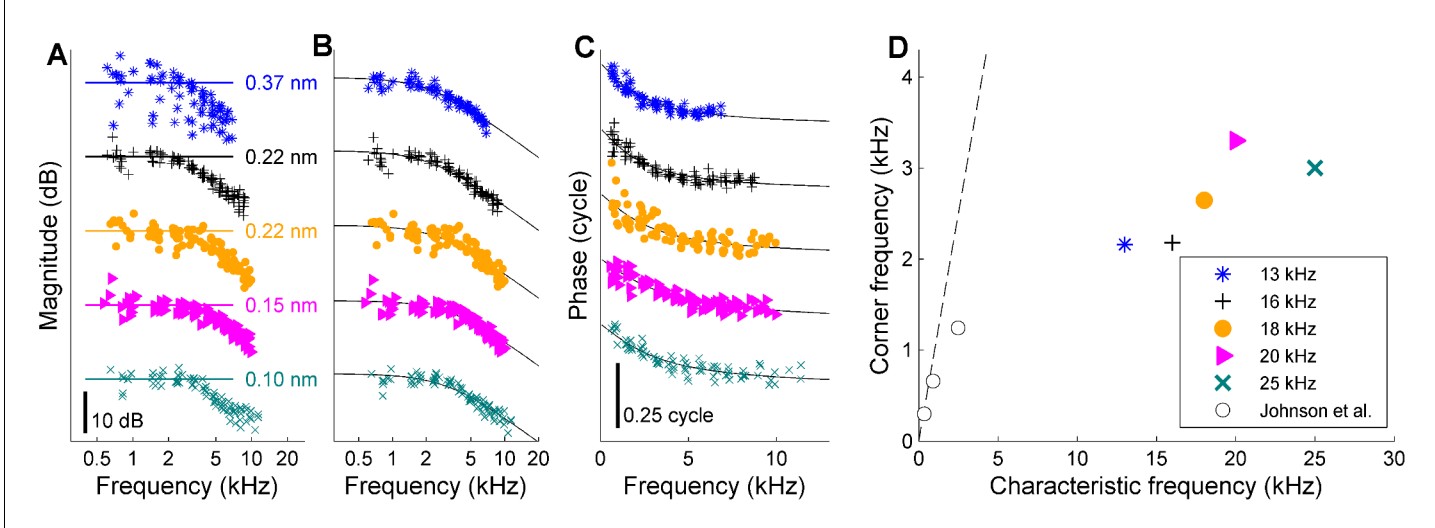

**Figure 4.** Low-pass filtering and corner frequencies of OHCs. (**A**) DP2 magnitude versus frequency measured across different cochleas at different CFs. Stimuli, 10–12 component zwuis not exceeding CF/2; component intensities 55–65 dB SPL, optimized to equalize OHC input. Individual curves are offset to avoid overlap; labeled straight lines indicate absolute displacement. CFs, see legend of panel D. Second harmonics were corrected for the 6-dB combinatorial deficiency. (**B**) The same magnitudes corrected *post-hoc* for the effects of residual magnitude inequality in the effective input (see *Appendix 1—figure 3*). Black lines, first-order low-pass filters fitted jointly to the magnitude and phase data of each recording. (**C**) Companion phase data: the difference between the recorded DP2 phases and the predictions obtained by adding or subtracting the primary phases of the OHC input (Appendix 1, *Equation 3B*). *Black lines*, phase curves of the fitted low-pass filters. (**D**) Corner frequencies from the fits versus CF. Symbols as in panels A-C. Open circles reproduce in vitro data from *Johnson et al. (2011)*. Dashed line, unity line (corner frequency = CF). Explained variance of the fits (in order of increasing CF): 90%, 91%, 80%, 85%, 87%. When omitting the correction for residual scatter, and instead using the raw magnitudes from panel A, the estimates of the corner frequencies were lower by 1% to 10% (mean, 7%) compared to the estimates based on the corrected magnitudes. Captions of source Data.

DOI: https://doi.org/10.7554/eLife.47667.012

The following source data and figure supplement are available for figure 4:

**Source data 1.** MATLAB binary file containing the data shown in *Figure 4*.
DOI: https://doi.org/10.7554/eLife.47667.014
**Figure supplement 1.** Sensitivity of the cochleae.
DOI: https://doi.org/10.7554/eLife.47667.013

receptor potential of 5.7 µV at the peak of traveling wave. At the slightly more basal location where cycle-by-cycle amplification is assumed to start, it is ~1 µV. Inspection of the in vivo OHC recordings in guinea pig of *Cody and Russell (1987)* yield an 3.6-µV AC component at CF for a 17-kHz tone near the behavioral threshold, corresponding to ~0.6 µV at the spatial onset of the putative amplification. Even if these minute variations in the membrane potential could evoke a significant electromotile response, such a motile feedback is unlikely to improve sensitivity because of its expected poor signal-to-noise ratio (*van der Heijden and Versteegh, 2015a*).

Overall our data suggest that OHCs and IHCs have similar properties, namely, considerable rectification (*Pappa et al., 2019*) and a corner frequency not exceeding a few kilohertz. Thus, just like in high-frequency IHCs, the receptor potential of high-frequency OHCs is expected to mainly follow the envelope of the waveform that stimulates their hair bundles. In this sense both IHC and OHCs operate as envelope detectors. We therefore propose that OHC motility does not provide cycle-by-cycle feedback, but rather modulates sound-evoked vibrations (*Cooper et al., 2018*; *van der Heijden and Versteegh, 2015b*). In this scenario the dynamic range compression in the cochlea is based on an automatic gain control system (*van der Heijden, 2005*) in which the degree of OHC depolarization determines the gain. The spatial confinement of the motile response to the OHC/Deiters' cell region presents another challenge to the prevailing theory that OHCs drive basilar membrane motion directly. It rather suggests that electromotility controls the local coupling between OHCs and Deiters' cells in a parametric fashion, perhaps dynamically adjusting the amount of dissipation in the Deiters' cell layer. This fundamentally different view of the function of OHCs has great

consequences for the experimental study of their role in hearing loss and the origins of the vulnerability of cochlear sensitivity. As to theoretical work, it is important that models of cochlear function, whether invoking cycle-by-cycle feedback or not, incorporate the findings of the present study.

## Materials and methods

### Overview
The materials and methods employed in this study are summarized below. More extensive details are provided elsewhere (*Cooper et al., 2018*).

Sound evoked vibrations were recorded from the ossicles and cochlear partitions of deeply anesthetized female gerbils (n = 27, weight = 53–75 g). Spectral-domain optical coherence tomography (SD-OCT) measurements were made from the first turn of the intact cochlea, under open-bulla conditions – optical access to the partition being provided through the transparent round window membrane. The hearing thresholds of the animals were assayed using tone-evoked compound action potential (CAP) measurements from a silver electrode placed on the wall of the basal turn of the cochlea.

### Animal preparation
Animals were anesthetized using intraperitoneally injected mixtures of ketamine (80 mg/kg) and xylazine (12 mg/kg). Supplementary (1/4) doses of the same mixture were administered at intervals of 10–60 min to maintain the anesthesia at surgical levels throughout subsequent procedures. All experiments were performed in accordance with the guidelines of the Animal Care and Use Committee at Erasmus MC (protocol AVD101002015304).

### OCT vibrometry
An SD-OCT system (Thorlabs Telesto TEL320C1) was used for interferometric imaging and vibration measurements. Cross-sectional (B-scan) and axial images (A-scans and M-scans) were triggered externally using TTL pulses phase-locked to the acoustic stimulation system (Tucker Davies Technologies system III) at a sampling rate 111.6 kHz. The theoretical resolution of the OCT system was ~3.5 µm across a 3.5 mm depth-of-field (i.e., z-range), but the optics of our recording system (a Mitotoyu IR imaging lens with an NA of 0.055) introduced an axial point spread function of ~6 µm FWHM and a lateral resolution (in the xy plane) of 13 µm (all assessed in air, with a refractive index of 1; corresponding intracochlear measurements should scale inversely with the refractive index of perilymph, which we assumed to be 1.3). The linear operating range of the OCT system was >500 µm. The amount of light incident on the cochlea was ~3.7 mW. The sensitivity of the A-scan's phase-spectra to vibration permitted measurement noise-floors that ranged from ~30 pm/$\sqrt{}$Hz in the cochlea down to ~3 pm/$\sqrt{}$Hz in the middle ear.

The OCT's measurement beam was not aligned with any of the cochlea's principal anatomical axes. The vibration measurements that we made should therefore be sensitive to structural movements in all three of the cochlea's principal dimensions (radial, transverse, and longitudinal). Specifically, in all recordings used for this study, the measurement beam pointed toward scala vestibuli, toward the apex of the cochlea, and away from the modiolus.

When mapping vibrations across the width of the cochlea partition (cf. *Figure 1C*, *Figure 1—figure supplement 2*), measurements were spaced at intervals of between 6 and 12 µm in the xy-plane.

### Basic response analysis
Responses were analyzed by Fourier transformation of the vibration waveforms derived from contiguous groups of 3 pixels in each M-scan, where each pixel covers a depth of ~2.7 µm in the fluid-filled spaces of the cochlea, and ~3.5 µm in the air-filled spaces of the middle-ear. The statistical significance of each response component was assessed using Rayleigh tests of the component's phase stability across time (*Cooper et al., 2018*).

## Acoustic stimulation

Acoustic stimuli were tailored to fit the nature of each experiment, as described below. Each stimulus was coupled into the exposed ear-canal using a pre-calibrated, closed field sound-system. Stimuli were generally presented for 12 s, with inter-stimulus intervals ~ 1 min.

Broad-band multi-tone 'zwuis' complexes (*van der Heijden and Joris, 2003*) were used to determine the characteristic frequency and sensitivity functions of each recording site (e.g. see *Figure 4— figure supplement 1*). Each broad-band stimulus had 43 spectral components, spanning from 0.4 to 30 kHz with an average spacing of 705 Hz. The components all had equal amplitudes, with levels expressed in decibels *re:* 20 μPa (i.e., dB SPL), but stimulus phase was randomized across frequency.

The unique property of a zwuis stimulus is that the frequencies of all of its primary components, and all of its potential inter-modulation distortion products up to the third order, are unambiguous. This means that all of the second-order distortion products (i.e. DP2s) studied in this paper can readily be attributed to a unique pair of spectral 'parents' (see Appendix 1).

Narrow-band zwuis stimuli were used to simplify the analysis and interpretation of DP2 spectra. They consisted from 10 to 15 components, ranging from few hundred hertz to at least one octave below the characteristic frequency of the recording side. The first presentation of each narrow-band stimulus had equal primary amplitudes, but their relative amplitudes were adjusted during subsequent presentations (fixing the average magnitude in dB SPL) in order to equalize the input to the OHCs. This procedure is described in the Appendix 1.

## Acknowledgements

This work was supported by the Netherlands Organization for Scientific Research, ALW 823.02.018, and an EU Horizon 2020 Marie Skłodowska-Curie Action Innovative Training network, H2020-MSCA-ITN-2016 [LISTEN - 722098].

## Additional information

### Funding

| Funder | Grant reference number | Author |
| --- | --- | --- |
| Horizon 2020 | H2020- MSCA-ITN-2016 [LISTEN - 722098] | Anna Vavakou |
| Nederlandse Organisatie voor Wetenschappelijk Onderzoek | ALW 823.02.018 | Nigel P Cooper |

The funders had no role in study design, data collection and interpretation, or the decision to submit the work for publication.

### Author contributions

Anna Vavakou, Data curation, Software, Validation, Investigation, Visualization, Methodology, Writing—original draft; Nigel P Cooper, Supervision, Validation, Investigation, Methodology, Writing—review and editing; Marcel van der Heijden, Conceptualization, Resources, Data curation, Software, Formal analysis, Supervision, Funding acquisition, Investigation, Visualization, Methodology, Project administration, Writing—review and editing

### Author ORCIDs

Marcel van der Heijden (iD) https://orcid.org/0000-0002-3876-1257

### Ethics

Animal experimentation: Experiments were performed in accordance with the guidelines of the Animal Care and Use Committee at Erasmus MC, which approved all protocols. Protocol number: AVD101002015304.

Decision letter and Author response
Decision letter https://doi.org/10.7554/eLife.47667.022
Author response https://doi.org/10.7554/eLife.47667.023

## Additional files

**Supplementary files**

• Transparent reporting form
DOI: https://doi.org/10.7554/eLife.47667.015

**Data availability**

Source data files have been provided for Figures 1 and 3, and 4.

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

## Appendix 1

DOI: https://doi.org/10.7554/eLife.47667.016

### Reduction of DP2 scatter by equalizing the effective OHC input

Here we describe the computational procedure leading to the reduction of magnitude scatter in the DP2 spectra (*Figures 3* and *4*). The computations are illustrated using the example primary spectrum of *Figure 2C* with its unequal primary magnitudes.

In a zwuis tone complex the primary frequencies $f_1 \ldots f_N$ are chosen such that all possible combinations $f^+_{km} = f_k + f_m$ ($k \geq m$) and $f^-_{km} = f_k - f_m$ ($k > m$) are distinct and never coincide with any primaries. This means that all the $N^2$ second-order distortion products (DP2s) generated by rectifying the stimulus are different, and can be uniquely traced back to their 'parent' primaries. *Figure 2C* of the main text shows the DP2 spectrum obtained by half-wave rectification of a 12-tone zwuis stimulus with unequal linear amplitudes $A_1 \ldots A_{12}$. To good approximation, and up to a common scale factor, the amplitude of a DP2 component at frequency $f_k \pm f_m$ ($k > m$) is equal to the product $A_k A_m$ of the parent amplitude (*van der Heijden and Joris, 2003*) whereas the amplitude of second harmonics at $2f_k$ equals $\frac{1}{2}A^2_k$ (*Meenderink and van der Heijden, 2010*)

*Appendix 1—figure 1A* shows a numerical test of the amplitude approximation. The unequal-amplitude stimulus shown in *Figure 2C* of the main text, was rectified (but not low-pass filtered) and the DP2 spectrum was extracted, corrected for the 6-dB combinatorial effect, and compared against the bilinear prediction. The predictions are accurate with a fraction of a dB except for the very weakest DP2s, which are up to 1.2 dB lower than predicted. (This deviation stems from the imperfect approximation of half-wave rectification in terms of a second-order distortion.) The prediction for the phase of a DP2 at $f_k \pm f_m$ is $\varphi_k \pm \varphi_m$, where $\varphi_1 \ldots \varphi_N$ are the primary phases (*Meenderink and van der Heijden, 2010*). *Appendix 1—figure 1B* shows the test of the prediction of DP2 phases from the primary phases using the same stimulus as in *Appendix 1—figure 1A*. The phase predictions are accurate to within 0.0025 cycle.

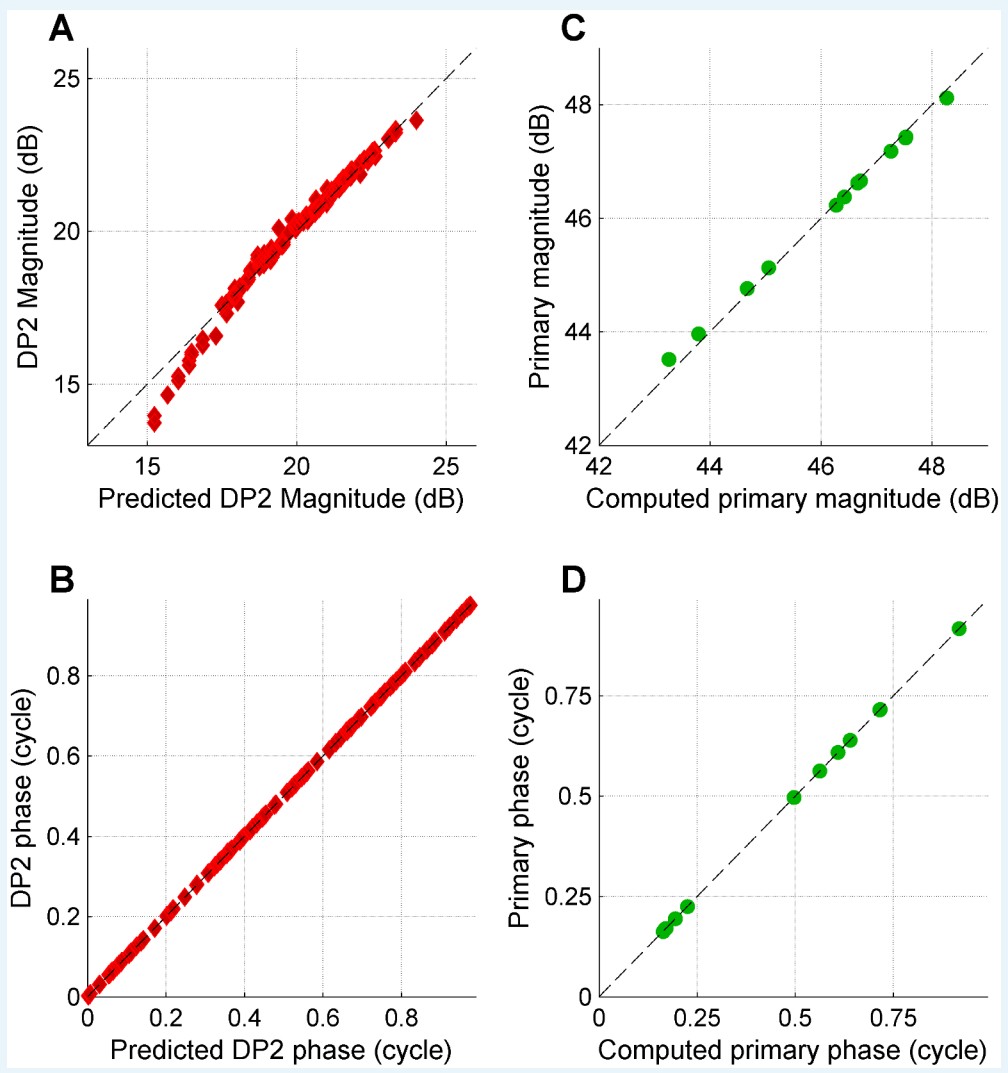

**Appendix 1—figure 1.** Predicting the DP2 spectrum from a primary spectrum and *vice versa*; no low-pass filtering. (**A**) Actual DP2 magnitudes obtained by rectifying (not followed by filtering) the tone complex with unequal primary magnitudes shown in *Figure 2C*, left panel, plotted against the approximation of *Equation 1*. (**B**) As in (**A**), but now for the phase. (**C**) Retrieving the primary magnitudes from the DP2 spectrum by inverting (fitting) *Equation 1*. Actual primary magnitudes are plotted versus computed magnitude. (**D**) As in (**C**), but now for the phase.

DOI: https://doi.org/10.7554/eLife.47667.017

The 'forward prediction' (DP2s from primaries) illustrated in *Appendix 1—figures 1A,1B* is of less importance; it is the reverse procedure that is used to reconstruct from the measured DP2 data the underlying primary amplitudes and phases. Specifically, denoting the primary magnitudes by $M_k = 20 log A_k$, and denoting DP2 magnitudes of sum tones at $f_{km}^+ = f_k + f_m$ and difference tones at $f_{km}^- = f_k - f_m$ by $M_{km}^+$ and $M_{km}^-$, respectively, and analogously for the phase, we obtain

$$M_{km}^\pm = M_k + M_m \tag{1A}$$

$$\varphi_{km}^\pm = \varphi_k \pm \varphi_m \tag{1B}$$

Note that scatter in the DP2 magnitudes results from any variation of primary magnitudes results. This includes regular trends in the primary magnitudes such as the deviation of a single primary component (e.g. due to a dip in the middle-ear transfer) or a systematic roll-off. Such regular trends in the input give scatter throughout the DP2 spectrum because each primary component affects DP2 components at multiple frequencies.

Retrieving the $N$ primary magnitudes and phases (up to overall offsets) from the $N^2$ DP2 magnitudes and phases amounts to solving the overdetermined set of *Equation 1* in a least squares sense. Because *Equation 1* is linear this leads to a unique and stable solution, and the numerical implementation is straightforward and efficient (e.g. MATLAB left matrix division). For $N = 12$ (as in *Appendix 1—figure 1*), $2 \times 11$ unknowns are retrieved from $2 \times 144$ knowns. The overdetermined character makes the procedure accurate and robust. A test of the retrieval of the relative primary magnitudes and phases from the DP2 spectrum is shown in *Appendix 1—figures 1C, 1D*. The deviations in the retrieved magnitude are systematic but small ($\leq 0.25$ dB); the phase is retrieved to within 0.00025 cycle.

When the rectifier is followed by a filter (whether low-pass or not) having complex transfer function $H_\alpha$ ($f$), where $\alpha$ stands for a set of parameters that characterize the filter, *Equation 1* is extended to

$$M_{km}^{\pm} = M_k + M_m + 20log|H_\alpha\left(f_{km}^{\pm}\right)| \tag{2A}$$

$$\varphi_{km}^{\pm} = \varphi_k \pm \varphi_m + arg\left(H_\alpha\left(f_{km}^{\pm}\right)\right) \tag{2B}$$

In principle, *Equation 2* can be used to fit the DP2 data in a least-squares sense as before, now also incorporating the new parameters $\alpha$ to the fit (in addition to the primary magnitudes and phases). When fitting experimental data, however, it is unclear a priori what type of filter to anticipate. To accommodate a variety of possible filter shapes, we extended *Equation 1* by adding 7th-order polynomials (increasing the order did not change the results):

$$M_{km}^{\pm} = M_k + M_m + \sum_{n=1}^{7} \beta_n \left(f_{km}^{\pm}\right)^n \tag{3A}$$

$$\varphi_{km}^{\pm} = \varphi_k \pm \varphi_m + \sum_{n=1}^{7} \gamma_n \left(f_{km}^{\pm}\right)^n \tag{3B}$$

Like *Equation 1*, this model is linear in its fit parameters, so it leads to a unique solution (in a least squares sense). In *Equation 3A* the primary magnitudes $M_k$ and $M_m$ describe the 'within-band' scatter of DP2 magnitudes, and the polynomial describes the post-rectifier filter. Fitting *Equation 3* to the DP2 spectrum of *Figure 2C* (which includes the low-pass filtering) reproduces the primary spectrum accurately (*Appendix 1—figure 2*). The largest deviations are 0.3 dB and 0.07 cycle.

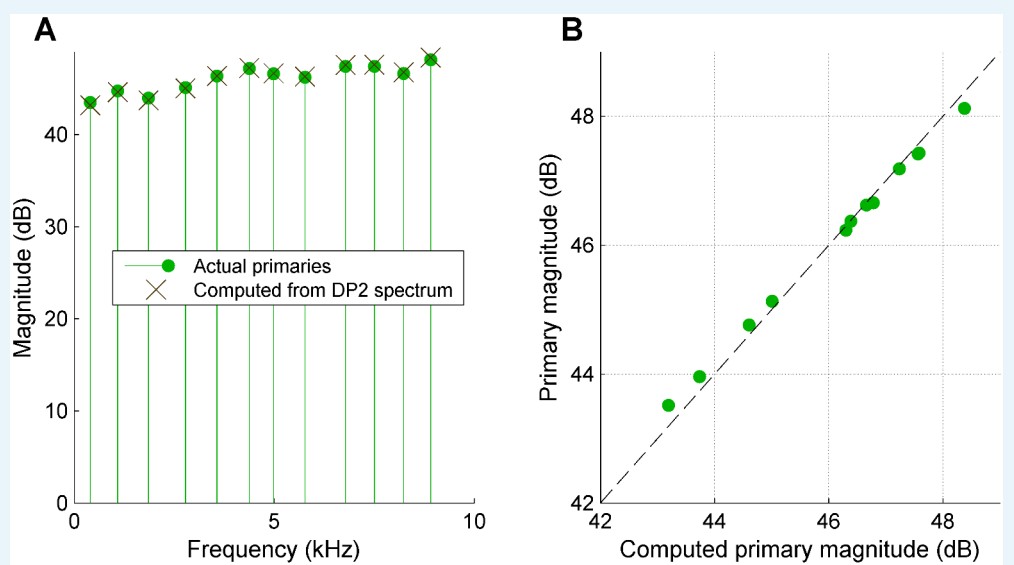

**Appendix 1—figure 2.** Retrieving the primary spectrum from the low-pass filtered DP2 spectrum. (**A**) A tone complex with non-equalized primary spectrum (*green circles*) was rectified and low-pass filtered at 2.5 kHz. From the resulting DP2 spectrum, the primary spectrum was reconstructed (*black Xs*). (**B**) Scatter plot comparing the actual primary spectrum entering the rectifier + low-pass filter scheme to the primary spectrum reconstructed from the DP2 spectrum at the output of the low-pass filter.

DOI: https://doi.org/10.7554/eLife.47667.018

Having retrieved the primary spectrum (whose lack of flatness causes the scatter of DP2s), we can assess the post-rectifier filter. This is illustrated in *Appendix 1—figure 3* for the magnitudes; the phase analysis is analogous. The actual DP2 spectrum of the rectified + low-pass filtered waveform is shown in *Appendix 1—figure 3A*. Inserting the retrieved primary magnitudes (shown in *Appendix 1—figure 2A*) into *Equation 1* yields the predicted unfiltered DP2 spectrum (*Appendix 1—figure 3B*). This isolates the scatter. Subtracting the scatter from the actual DP2 spectrum retrieves the effect of the filter (*Appendix 1—figure 3C*). This is the DP2 spectrum 'corrected for scatter' (for the experimental data this scatter-corrected version of the magnitudes is shown in *Figure 4B* of the main text). It clearly reproduces the first-order low-pass filter used to generate the DP2 spectrum, which is shown in *Appendix 1—figure 3C* for reference. Note that up to this point nothing in the fitting procedure has presumed a low-pass filter; the polynomial terms of *Equation 3* are agnostic in this respect.

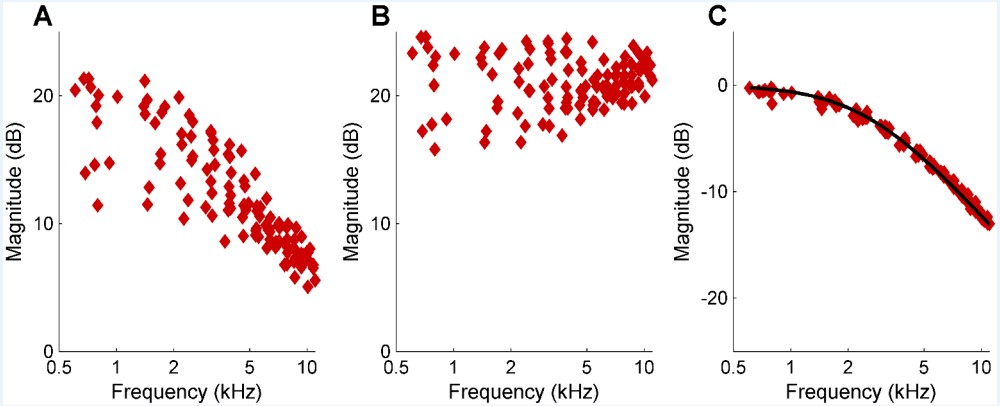

**Appendix 1—figure 3.** Computational separation of DP2 scatter and filter effect. (**A**) DP2 spectrum obtained by rectifying and low-pass filtering a zwuis multitone waveform having

unequal primary amplitudes. (**B**) Magnitude scatter in the DP2 spectrum of panel (**A**), computed by inserting the retrieved primary magnitudes into **Equation 1**. (**C**) The effect of the low-pass filter isolated by subtracting the scatter contribution of panel (**B**) from the DP2 spectrum in (**A**). For reference, the gain curve of actual filter that was used to generate the DP2 spectrum (first-order low-pass; corner frequency 2.5 kHz) is also shown (*black line*).
DOI: https://doi.org/10.7554/eLife.47667.019

Theoretically, the above computational procedure is all that is needed to isolate and estimate the filter contribution to the DP2 spectrum (**Appendix 1—figure 3C**). In the experiments, we went further and used the retrieved OHC input to adapt the acoustic stimulus aimed at equalizing the OHC input and reducing the DP2 scatter (this step was iterated if necessary). We had two reasons for doing so. (1) Equalized OHC input spectra yield higher numbers of Rayleigh-significant DP2 components, thus richer datasets. This can be understood from the limited dynamic range of the measurements: if the DP2 magnitude scatter is too large, the weaker DP2s will drop below the noise floor. (2) Scatter reduction by adapting the stimulus is a powerful way of interrogating the system. The simple rectifier + filter scheme predicts that an adjustment of the relative SPLs of $N$ primaries (i.e. using $N$-1 degrees of freedom) reduces the scatter in as many as $N^2$ distortion components in the raw data. This is a strong prediction (recall that $N \geq 10$ in our recordings), and to see it happening in the real data (**Figures 3** and **4**) confirms that the simple scheme is an adequate description of the DP2 spectra generated by the OHCs.

Fitting a first-order low-pass filter to the scatter-corrected DP2 spectrum (the last analysis step) was done jointly to the magnitude (**Figure 4B**) and phase data (**Figure 4C**). Consider a DP2 having a magnitude of $M$ dB and a phase of $\varphi$ cycles. In the complex spectrum of the response it is represented by a component $Z = 10^{M/20} e^{2\pi i \varphi} = e^{(ln10/20)M + 2\pi i \varphi}$, or

$$ lnZ = (ln10/20)M + 2\pi i \varphi \qquad (4) $$

an expression that exposes the common logarithmic nature of $M$ and $\varphi$, and provides the natural conversion factor between the two in the form of the ratio $\Theta = 2\pi/(ln10/20) \simeq 54.6$ dB/cycle (this conversion factor would be unity if magnitude and phase were expressed in nepers and radians, respectively). The joint fit of magnitude data $M$ (in dB) and phase data $\varphi$ (in cycles) then amounts to minimizing the sum of squares

$$ \chi^2 = \sum (M_{\mathrm{data}} - M_{\mathrm{model}})^2 + \Theta^2 \sum (\varphi_{\mathrm{data}} - \varphi_{\mathrm{model}})^2 \qquad (5) $$

This expression was minimized to produce the fits shown as black lines in **Figure 4B and C**.

## Appendix 2

DOI: https://doi.org/10.7554/eLife.47667.016

### Estimate of the AC receptor potential of OHCs near threshold

In this Appendix estimates are made of the AC receptor potential in OHCs in response to high-frequency tones near behavior threshold. Specifically, the AC receptor potential evoked by a 17-kHz tone at 5 dB SPL in guinea pig are derived from the in vivo OHC recordings of *Cody and Russell (1987)*, and the AC receptor potential evoked by a 16-kHz, 5-dB-SPL tone in gerbil is estimated by combining recordings of basilar-membrane vibrations with electrophysiological data.

Figure 6B of *Cody and Russell (1987)* shows a 160-μV AC receptor potential for a 17-kHz tone at 15 dB SPL. These data were corrected for the membrane time constant, for which the authors used 6 dB per octave above 1200 Hz. Thus the raw AC component $V_{AC}$, corrected for equipment filtering but not for the membrane time constant, was $160 \times 1200/17000 = 11.3$ μV. In this low-SPL regime the growth is linear, leading to an estimate of $V_{AC} = 3.6$ μV for a 5-dB-SPL tone. These recordings originate from the 17-kHz location, where the wave is assumed to have been amplified. At the spatial onset of the putative amplification, basal to the best 17-kHz location, the excitation is smaller by ~15 dB (see below), leading to $V_{AC} = 0.6$ μV at the spatial onset of compressive growth.

The AC receptor potential of OHCs measured in vivo saturates with low-frequency (200 Hz) stimulation at ~95 dB SPL (Fig. 11 of *Dallos, 1986*) and this matches the saturation of the cochlear microphonic potential at the round window, which is dominated by basal OHCs (*Dallos and Cheatham, 1976*), observed with 95-dB-SPL tones at 200 Hz (Fig. 4 of *Patuzzi et al., 1989*). In gerbil, a 95-dB-SPL tone at 200 Hz evokes a basilar membrane displacement of ~300 nm. Combining these two findings, the Boltzmann function relating BM displacement $d_{BM}$ to the mechano-transducer channel conductivity $G_{MT}$,

$$G_{MT}(d_{BM}) = G_{max}/\left(1 + e^{-\beta d_{BM}}\right) \tag{1}$$

has $\beta \approx 0.01$ nm$^{-1}$. ($G_{max}$ is the maximum value with all transducer channels open.) Assuming that the resting value $G_{MT,0}$ of the MT conductivity equals $G_{max}/2$ (this is where the Boltzmann function is steepest, maximizing the AC component of the receptor current), the fractional conductivity variation (modulation $m$) for small BM displacements equals

$$m = \Delta G_{MT}/G_{MT,0} = \frac{\beta}{2} d_{BM} \tag{2}$$

The behavior threshold for 16-kHz tones in gerbil is 5 dB SPL (*Ryan, 1976*), and at the 16-kHz location of the sensitive gerbil cochlea, this evokes a BM displacement of $d_{BM} \approx 0.1$ nm, corresponding to $m \approx 5 \times 10^{-4}$. This is the value at the peak excitation following the putative amplification. The actual SPL-dependent (physiologically vulnerable) amplitude growth occurs basal to the peak. In the 16-kHz region the amplitude growth amounts to ~15 dB at the lowest SPLs (Fig. 3A of *Ren, 2002*). Thus at the spatial onset of the putative amplification of the near-threshold 16-kHz tone, $d_{BM} \approx 0.018$ nm, corresponding to $m \approx 9 \times 10^{-5}$.

The AC receptor potential follows from a straightforward linearization of the equivalent OHC circuit (e.g., Fig. 6B of *Johnson et al., 2011*),

$$V_{AC} = m(E_K + E_{EP})/\left[\left(1/G_K + 1/G_{MT,0}\right)\left(G_K + G_{MT,0} + i\omega C\right)\right] \tag{3}$$

using a total basolateral membrane capacitance $C = 5$ pF (neglecting the apical membrane capacitance); deriving the basolateral membrane resting conductance $G_{K,0}$ from our corner frequency: $G_{K,0} = C\omega_0 = 2\pi C f_{corner}$, with $f_{corner} = 2200$ Hz (our *Figure 4*); the K$^+$ reversal potential of $E_K = -75$ mV; the resting value of MT conductivity $G_{MT,0} = 75$ nS (Fig S1 of *Johnson et al., 2011*); an endocochlear potential $E_{EP} = 90$ mV; and the angular stimulus

frequency $\omega = 2\pi f$. Applying *Equation 3* to the values of fractional MT conductivity $m$, we obtain $V_{AC} \approx 5.7$ µV at the peak of the wave and $V_{AC} \approx 1.0$ µV at the spatial onset of its nonlinear growth.

