## [Decision Letter]

Thank you for submitting your article "The frequency limit of outer hair cell motility measured in vivo" for consideration by *eLife*. Your article has been reviewed by three peer reviewers, including Tobias Reichenbach as the Reviewing Editor and Reviewer #1, and the evaluation has been overseen by Andrew King as the Senior Editor. The following individuals involved in review of your submission have agreed to reveal their identity: Elizabeth Olson (Reviewer #2); Thomas Risler (Reviewer #3).

The reviewers have discussed the reviews with one another and the Reviewing Editor has drafted this decision to help you prepare a revised submission.

Summary:

This is a well-presented and illustrated paper that addresses an important issue that has been controversial in the field of cochlear mechanics for years: is OHC prestin-based motility capable of amplifying the input signals on a cycle-by-cycle basis at auditory frequencies, up to several tens of kHz and above 100 kHz in some species? As the authors correctly point out, accurate in-vivo data acquired at higher frequencies are crucial to establish a clear cut answer to that essential question. The authors present convincing in vivo data that show that the corner frequencies of the outer hair cells are comparatively low, about a few kHz.

Essential revisions:

1) A major issue concerns the interpretation of the results, in particular the authors' conclusion that the low corner frequencies rule out cycle-by-cycle amplification by OHCs in the cochlea. For example, in Figure 3 the stimulus level is ~ 55 dB and the DPs are ~ 0.3 nm at 2 kHz (a frequency below the corner frequency) and ~ 0.1 nm at 10 kHz. From the literature, the response to primaries would be ~ 0.1-1 nm at this level (Cooper et al., 2018, Figure 6 panels c-f). Presumably the electromotility at the primaries is larger than the DP electromotility (because the primaries are bigger than the DPs), which means the expected primary electromotility response is roughly equal to the actual primary responses even with the corner frequency taken into account. This example is meant to emphasize that the question of what is "big enough" is a quantitative issue. How big does electromotility need to be in order to work effectively on a cycle-by-cycle basis? Computational models show that amplification at high frequencies can potentially work with comparatively slow OHC electromotility, in particular when combined with active hair-bundle motility (see, e.g., Maoileidigh and Jülicher, JASA 2010; Meaud and Gross, Biophysics. J. 2011). The low corner frequencies that the authors find here are clearly relevant for modelling, but the specific implications are less clear. Please discuss these issues.

2) Could the authors explain and justify their choice of restricting the range of both primaries and DP2s to frequencies not exceeding half of the characteristic frequency (why not another fraction)?

3) Please show at least one full frequency response to illustrate that these were healthy cochleae. Please show these data on a nm scale (ie, not normalized to stapes). As you note, you don't know what the input to the OHC stereocilia is (where the nonlinearity is that will produce the DPs), but a first approximation will be the motion of the primaries so please show that – show the parent f1,f2 in addition to the DPs.

4) The phase reference is not obvious, please include that info when plotting phase. (We assume that it is ph2-ph1, ph2+ph1 of the OHC primary motion response.) Also, depending on which side of the IO curve the system is operating on, the sign of the DP would be different, which would result in a 180 phase difference. Was evidence of that ever seen in preparations? (For example, see Figure 2 of Brown et al., JASA 125, 2009.)

5) Where does the scatter in the amplitudes of the primary components of the zwuis stimulus come from? Is it already present in the sound stimulus, or does it reflect irregularities in how the sound propagates into the cochlea? It would also be good to see if this scatter is robust for a given ear, that is, whether one obtains the same scatter when the sound system is detached and then re-attached to a given ear.

6) Is correction 3 actually required? We think that it would be good to compare the results obtained from the corrected data to those obtained without correction 3, since the latter results might be similarly clean. In particular, it appears that the phase is not affected by the primary scatter and that is favorable to your argument, and in many of your examples the scatter is not so large as to obscure the low-pass trend of the amplitude even without correction 3.

[Editors' note: further revisions were requested prior to acceptance, as described below.]

Thank you for resubmitting your work entitled "The frequency limit of outer hair cell motility measured in vivo" for further consideration at *eLife*. Your revised article has been favorably evaluated by Andrew King (Senior Editor) and three reviewers, one of whom is a member of our Board of Reviewing Editors.

The manuscript has been improved but there are some remaining issues that need to be addressed before acceptance, as outlined below:

Abstract: "exceeds the electrical limits" is vague and sounds worse than the reality. It's better to be quantitative and write something like "are attenuated by a factor of ~ 7".

Introduction section: “Again, in vivo data are missing” Please replace "missing" with "minimal" since in vivo data do exist and are referred to later.

Results and Discussion paragraph four: For clarity, please write "BM motions and calculated OHC input"

Results and Discussion paragraph seven: A reference to "In-vivo impedance of the gerbil organ of Corti at auditory frequencies." Biophysical Journal 97: 1233 – 1243" would provide support for the statement that the OOC impedance is stiffness dominated.

Results and Discussion paragraph eight: Johnson et al. predicted the electrical corner frequency, whereas your measurements are of the corner frequency due to electrical + any mechanical low-pass filtering. (You alluded to the possibility of mechanical filtering in the Introduction.) Thus a direct comparison with Johnson et al. needs qualification here.

Paragraph nine: Please either say "much stronger at much higher" or "stronger at higher" (we suggest the latter.)

Final paragraph: Please delete "clear-cut" – the reader should decide this on their own.

Figure 2—figure supplement 2. To make this figure easier to access, we suggest adding to the caption after the first sentence something along the lines of, "In A all the DP components relating to the lowest primary are shown. In B, that lowest primary is excluded and all the DP components relating to the new lowest primary are shown, and so on through panel K."

Results and Discussion paragraph three: This relates to our previous comment on the first version. The authors correctly addressed the combinatorial effect, but kept the larger or equal k > = m in the parenthesis. It seems that it should be k>m (and not equal), otherwise we fall back to the 2*f_k_* case.

Appendix 1—figure 2: Regarding our previous comment on this figure, the authors acknowledged that there was a typo referring to Figure 3C and not Figure 2C (after Equation 3B). They have however omitted to correct for that in the revised manuscript. Please correct this.

---

## [Author Response]

Summary:This is a well-presented and illustrated paper that addresses an important issue that has been controversial in the field of cochlear mechanics for years: is OHC prestin-based motility capable of amplifying the input signals on a cycle-by-cycle basis at auditory frequencies, up to several tens of kHz and above 100 kHz in some species? As the authors correctly point out, accurate in-vivo data acquired at higher frequencies are crucial to establish a clear cut answer to that essential question. The authors present convincing in vivo data that show that the corner frequencies of the outer hair cells are comparatively low, about a few kHz.

We thank the Editor and reviewers for their careful comments and helpful suggestions. Before addressing the individual comments, we would like to clarify a general methodological aspect that is relevant to several of them. The central results of our study (Figure 4) were obtained exclusively from the distortion (DP2) spectrum. Only the DP2 part of the response could be ascribed with confidence to electromotility. In contrast, the *linear* response part recorded in the OHC region is an unknown mix of passive (non-vulnerable) and motile contributions. Thus a priori one may not expect the linear component of OHC motion to be a good proxy for the input to the OHCs that shapes its motile response. In fact, as shown below, BM motion resembles OHC input more than does OHC motion itself.

Essential revisions:1) A major issue concerns the interpretation of the results, in particular the authors' conclusion that the low corner frequencies rule out cycle-by-cycle amplification by OHCs in the cochlea. For example, in Figure 3 the stimulus level is ~ 55 dB and the DPs are ~ 0.3 nm at 2 kHz (a frequency below the corner frequency) and ~ 0.1 nm at 10 kHz. From the literature, the response to primaries would be ~ 0.1-1 nm at this level (Cooper et al., 2018, Figure 6 panels c-f). Presumably the electromotility at the primaries is larger than the DP electromotility (because the primaries are bigger than the DPs), which means the expected primary electromotility response is roughly equal to the actual primary responses even with the corner frequency taken into account. This example is meant to emphasize that the question of what is "big enough" is a quantitative issue. How big does electromotility need to be in order to work effectively on a cycle-by-cycle basis? Computational models show that amplification at high frequencies can potentially work with comparatively slow OHC electromotility, in particular when combined with active hair-bundle motility (see, e.g., Maoileidigh and Jülicher, JASA 2010; Meaud and Gross, Biophysics. J. 2011). The low corner frequencies that the authors find here are clearly relevant for modelling, but the specific implications are less clear. Please discuss these issues.

For the relative magnitudes of DP2s and linear response components, see the reply to comment #3 and the new Figure 2—figure supplement 2. Please note that the vibration amplitudes recorded in the OHC/Deiters’ cell region in response to 55-dB-SPL tones in Figure 6d of Cooper et al., 2018, are much higher than the ~0.1-1 nm mentioned in the comment: they are 4.2 nm at 2.3 kHz and 1.4 nm at 9.3 kHz. Even so, the direct comparison of amplitudes between linear components and distortion products is of limited use because overall DP2 magnitude (but not the relative DP2 amplitudes) depends on the unknown degree of nonlinearity. In the above comment and elsewhere in the review the assumption seems to be made that OHC vibrations are primarily motility-driven in the first place. As shown below this assumption is contradicted by the data, but in any case we object (on logical grounds) against using this premise when reasoning about the potency of motility to drive motion in the cochlea.

We agree that a quantitative analysis may be helpful. Using a refinement of the derivation in Versteegh and Van der Heijden (2015), we estimated the AC receptor potential evoked by a 16-kHz tone at 5 dB SPL, i.e., at the behavioral threshold of the gerbil. The calculations are presented in the new Appendix 2. In the penultimate paragraph of the main text we now briefly discuss the outcome in the context of cycle-by-cycle feedback.

Apart from these quantitative aspects, there is the scientific context created by the 35-year-long debate on this topic. That corner frequencies much smaller than CF “spell trouble” for somatic cycle-by-cycle amplification is a view expressed in a large body of literature. For instance, Ashmore, 2011, writes “In the intact cochlea, however, the electrical filtering effect of the cell membrane, effectively possessing an electrical time constant = R_m_C_m_, would reduce potential changes to negligible levels at any significant acoustic frequencies.” It is against this background that we discuss our result as being problematic for cycle-by-cycle amplification, and propose potential alternatives for the role of somatic motility. In the new text in the penultimate paragraph we touch upon this context by addressing previous proposals aimed at circumventing the RC problem.

2) Could the authors explain and justify their choice of restricting the range of both primaries and DP2s to frequencies not exceeding half of the characteristic frequency (why not another fraction)?

Explanation: In order to isolate the effect of DP generation one must stay well away from the effects of slow propagation of the primaries and the DPs. Since any deviation in primary amplitude and phase will be doubled in the DP2 components, it is wise to be on the safe side. The iterative equalization method is based on the computation of the effective input from the previous recording. This computation depends on the choice of the frequency range of both primaries and DP2s, so we had to fix the choice prior to collecting the bulk of the data. From the pilot experiments we observed that CF/2 was certainly safe enough and we stuck to it for consistency. Note also that using primaries above CF/2 results in their sum DP2s being above CF, which typically makes them too weak to record. Such an a priori waste of DP2 components spoils the redundancy needed for computing the effective OHC input.

Justification: DP2s do propagate and in doing so accumulate phase and amplitude changes that are unrelated to the local parent primaries (Figure 2—figure supplement 1). The choice of CF/2 led to an accurate and reproducible determination of the effective input spectrum, which in turn enabled us to iteratively reduce the DP2 scatter. The frequency range is large enough by a considerable margin to observe the first-order filtering and to fit the data (Figure 4). In all this, an important practical consideration is the time it takes to record, transfer the raw data from the measurement computer, perform the basic processing steps and compute the spectrum of the next stimulus in the adaptive process leading to equalization of the DP2 spectrum. The cycle is 15-20 minutes and we typically need 3 cycles. In vivo experiments that require sensitive cochleae allow little room for non-critical fine tuning.

In the text we clarified the CF/2 choice and its relation to the iterative method, including an explicit reference to Figure 2—figure supplement 1.

3) Please show at least one full frequency response to illustrate that these were healthy cochleae. Please show these data on a nm scale (ie, not normalized to stapes). As you note, you don't know what the input to the OHC stereocilia is (where the nonlinearity is that will produce the DPs), but a first approximation will be the motion of the primaries so please show that – show the parent f1,f2 in addition to the DPs.

This comment has two different aspects: (A) show the relation between DPs and linear response components; (B) show the sensitivity of the cochleae.

A) As requested, we provided a new figure (Figure 2—figure supplement 2) that combines linear components and DPs on a nm scale. These are responses to the main stimuli of the study (Figure 4, highest CF). While we indeed note in the manuscript that we cannot *directly* record the input to the OHCs, we do know it accurately from the analysis of DP2s (up to an overall scaling factor, see Appendix 1). The OHC input is also shown in Figure 2—figure supplement 2. Contrary to the expectation expressed in the reviewers’ comments, the vibrations in the OHC region are not a good approximation of the input of to the OHCs. In fact, BM motion (also shown in Figure 2—figure supplement 2) is a better proxy for the OHC input. This fits with Ter Kuile’s mechanism for hair-bundle excitation by the tilting of the tunnel of Corti following BM deflection. It is also consistent with the fact that suppression, which is mediated by OHCs, is accurately predicted by BM displacement over a wide frequency range (Cooper, 1996; Versteegh and van der Heijden, 2013). It also agrees with the observation made and discussed in Cooper et al., 2018, that *inner* hair cell excitation (as reflected by tuning in the auditory nerve) resembles basilar membrane motion more than hotspot motion. The new Figure 2—figure supplement 2 is cited, when introducing the computation of OHC input. The contrast between OHC input and OHC motion is mentioned, as is the resemblance of BM motion and OHC input. We do not elaborate on these findings here because they do not directly relate to the OHC corner frequency.

The expectation that OHC motion would reflect OHC input is expressed multiple times in the review. We also in Author response image 1 address this question in a complementary way. It compares two different equalization strategies: (1) equalizing linear OHC motion; (2) equalizing OHC input computed from the DP2 spectrum (as in the rest of the study). The scatter in the DP2 spectrum is much larger with (1) than with (2), confirming that OHC motion is a poor predictor of OHC input. This graph is not included in the manuscript because it is not directly related to the OHC corner frequency.

B) Cochlear sensitivity cannot be assessed using the main stimulus used in this study since it has no near-CF components. We therefore provide (Figure 4—figure supplement 1A) a set of responses to a wide range of frequencies (including CF) at multiple SPLs, recorded from the CF=18 kHz cochlea of Figure 4. They display the familiar compressive behavior of sensitive cochleae. For comparison Figure 4—figure supplement 1B shows 30-dB-SPL recordings of all five cochleae of Figure 4. The new Figure 4—figure supplement 1 is called out in the Materials and methods section.

4) The phase reference is not obvious, please include that info when plotting phase. (We assume that it is ph2-ph1, ph2+ph1 of the OHC primary motion response.) Also, depending on which side of the IO curve the system is operating on, the sign of the DP would be different, which would result in a 180 phase difference. Was evidence of that ever seen in preparations? (For example, see Figure 2 of Brown et al., JASA 125, 2009.)

Phase reference. Just like the magnitude data, the phase data shown in Figure 4 are exclusively derived from the DP2 spectrum. What is shown in Figure 4C (and explained in Appendix 1) is the difference between the measured DP2 phase and the DP2 phase predicted to occur after rectification (without filtering) of the OHC input having the computed phases. This prediction is indeed ph2-ph1, ph2+ph1, but the primary phases ph1, ph2 are taken from the effective OHC input. The phase reference is now mentioned in the caption of Figure 4C, including a reference to Appendix 1, Equation 3B.

Direction of rectification. The waveform in Figure 1B is representative of the “polarity” of the rectification in all the data in this study. We now mention in the caption what the positive direction in Figure 1B means in terms of the measurement beam, and in Materials and methods we now mention the relation between beam direction and each of the anatomical axes of the OoC. Again, however, we caution against overinterpretation. Information regarding IO curves can only be obtained by direct comparison of the input and the output, and as explained under #1 and #3, recorded OHC motion may not be identified with the output of electromotility. From the perspective of the desired IO analysis, OHC motion is “polluted” by non-motile-driven motion.

The link between phase offset and polarity of rectification is now pointed out in the caption of Figure 3C. In the caption of Figure 2C we now clarify low-frequency limit of the phase offset in terms of the direction of the rectifier.

5) Where does the scatter in the amplitudes of the primary components of the zwuis stimulus come from? Is it already present in the sound stimulus, or does it reflect irregularities in how the sound propagates into the cochlea? It would also be good to see if this scatter is robust for a given ear, that is, whether one obtains the same scatter when the sound system is detached and then re-attached to a given ear.

As shown in Author response image 2, detaching and re-attaching the ear coupler resulted in minor variations in stapes vibration (RMS deviation, 1 dB). We realize that our use of the term “scatter” has been a bit imprecise. If we define scatter as “distributed deviations from a regular pattern” then scatter in the DP2 spectrum results from *any* magnitude differences among primaries of the OHC input. This includes regular (“non-scattered”) trends in the OHC input such as the deviation of a single component (e.g. from a notch in the middle-ear transfer) as well as systematic roll-offs. Such regular trends in the input give scatter throughout the DP2 spectrum because each primary component affects DP2 components at multiple frequencies. We now point this out in Appendix 1 (following Equation 1), and replaced the term “scatter” by “unequal magnitudes” or equivalent whenever referring to the OHC input. We can think of three causes of non-equalized spectrum at the OHC input when using an equal-amplitude acoustic stimulus: (1) imperfections in the calibration; (2) middle ear transfer characteristics; (3) non-flatness of the intracochlear transfer from stapes to OHC bundle deflection. In each case, irregularities as well as systematic trends will result in scatter of the DP2 spectrum, and the range of the Dp2 scatter will be twice the range of the input magnitudes (see Appendix 1—figure 1). We now briefly mention these potential causes in the main text (description of Figure 2B) and Appendix 1 (following Equation 1).

**Author response image 2. respfig2:** 

6) Is correction 3 actually required? We think that it would be good to compare the results obtained from the corrected data to those obtained without correction 3, since the latter results might be similarly clean. In particular, it appears that the phase is not affected by the primary scatter and that is favorable to your argument, and in many of your examples the scatter is not so large as to obscure the low-pass trend of the amplitude even without correction 3.

When using the raw data of Figure 4A (instead of the corrected ones of Figure 4B), corner estimates are slightly lower than the reported ones, namely by 1 to 10% (mean, 7%). This is now mentioned in the caption. Still, the correction is necessary. Our method is valid on the grounds that any trend that cannot be explained by variation in the primaries magnitude is attributed to filter properties of the OHCs. Thus any scatter introduced by primary magnitude variation should be removed from our data.

[Editors' note: further revisions were requested prior to acceptance, as described below.]

The manuscript has been improved but there are some remaining issues that need to be addressed before acceptance, as outlined below:Abstract: "exceeds the electrical limits" is vague and sounds worse than the reality. It's better to be quantitative and write something like "are attenuated by a factor of ~ 7".

At this introductory stage of the Abstract, there is no factor of 7 yet (it is part of our findings). The sentence merely introduces the problem to be addressed; it contains no claims as to its gravity. We chose the phrasing “frequencies that exceed the electrical limits (i.e. corner frequencies) of their cell membranes” in order to briefly explain what a corner frequency is. We did so because the reviewers had asked us to make the Abstract more accessible to non-specialist readers.

We have kept the formulation as is, but we would be happy to change it to “frequencies that exceed the corner frequency of their cell membranes” if the Editor prefers this.

Introduction section: “Again, in vivo data are missing” Please replace "missing" with "minimal" since in vivo data do exist and are referred to later.

In vivo studies that used high-frequency current injection to evoke mechanical responses (e.g. Ren et al., 2016) have not yielded estimates of corner frequencies, the topic being reviewed here. We clarified this by replacing “in vivo data” by “in vivo estimates of the corner frequency of motility”

Results and Discussion paragraph four: For clarity, please write "BM motions and calculated OHC input"

Done.

Results and Discussion paragraph seven: A reference to "In-vivo impedance of the gerbil organ of Corti at auditory frequencies." Biophysical Journal 97: 1233 – 1243" would provide support for the statement that the OOC impedance is stiffness dominated.

We agree and have inserted the reference.

Results and Discussion paragraph eight: Johnson et al. predicted the electrical corner frequency, whereas your measurements are of the corner frequency due to electrical + any mechanical low-pass filtering. (You alluded to the possibility of mechanical filtering in the Introduction.) Thus a direct comparison with Johnson et al. needs qualification here.

We added “electrical” in “which predict electrical corner frequencies.” In the subsequent sentence we added “and minus 0.25-cycle phase asymptote” when addressing the first order lowpass character of our data. The next sentence is now expanded to “Comparison with the in vitrodata suggests that this dominant factor is the RC time of the cell membrane, which is fundamental to the operation of all biological cells.” We feel that the alternative explanation (mechanical filtering being the dominant factor), although theoretically possible, is too farfetched to warrant discussion. It would amount to denying well-known electrical properties of cells (even beyond Johnson et al’s bold extrapolations) with little functional implication or relevance, as it does not push the corner frequency of motility itself.

Paragraph nine: Please either say "much stronger at much higher" or "stronger at higher" (we suggest the latter.)

We changed it to “an even stronger attenuation at higher CFs than studied here”, underscoring that 17 dB is already a considerable attenuation (and alluding to the much higher CFs found in some species mentioned in the Introduction).

Final paragraph: Please delete "clear-cut" – the reader should decide this on their own.

Deleted.

Figure 2—figure supplement 2. To make this figure easier to access, we suggest adding to the caption after the first sentence something along the lines of, "In A all the DP components relating to the lowest primary are shown. In B, that lowest primary is excluded and all the DP components relating to the new lowest primary are shown, and so on through panel K."

We inserted this very helpful description exactly as suggested.

Results and Discussion paragraph three: This relates to our previous comment on the first version. The authors correctly addressed the combinatorial effect, but kept the larger or equal k > = m in the parenthesis. It seems that it should be k>m (and not equal), otherwise we fall back to the 2f_k_ case.

Apologies – now corrected.

Appendix 1—figure 2: Regarding our previous comment on this figure, the authors acknowledged that there was a typo referring to Figure 3C and not Figure 2C (after Equation 3B). They have however omitted to correct for that in the revised manuscript. Please correct this.

Apologies – now corrected.

To strengthen the requested quantitative analysis (Appendix 2), we added a rather straightforward estimate of the AC receptor potential at 17 kHz based on Cody and Russell’s, 1987, in vivo OHC recordings. The estimate is briefly mentioned in the penultimate paragraph of the main text.

We also spotted and corrected an error in the references (van der Heijden and Versteegh, 2015a).